# Effects of Leaving Amputated Ovaries Intra-Abdominally during Elective Bilateral Standing Laparoscopic Ovariectomy in Equids

**DOI:** 10.3390/ani11010232

**Published:** 2021-01-18

**Authors:** Ian F. Devick, Dean A. Hendrickson

**Affiliations:** Department of Clinical Sciences, Colorado State University, Fort Collins, CO 80523, USA; idevick@hotmail.com

**Keywords:** equine, laparoscopy, ovariectomy, endocrine, hormone, complications, behavior

## Abstract

**Simple Summary:**

When removing the ovaries of a mare via laparoscopic technique, there is potential to drop the ovary in the abdomen when trying to exteriorize it through the abdominal wall. If the ovary can no longer be seen with the laparoscope, then the procedure is converted to a flank incision and the ovary is identified and removed by hand. Having to convert the procedure negates the benefits of the minimal invasive laparoscopic procedure and increases the risk of post-operative complications. The objective of this study was to identify if amputated ovaries left in the abdomen during surgery would atrophy or if they would regain a blood supply and produce hormones. After surgery, the mare’s hormone values were at low levels and an improvement in all mare’s behavior and general herd dynamics was observed. While this study does not encourage to leave ovaries in the abdomen after amputation, we report no complications related to their voluntary release into the abdomen.

**Abstract:**

There is risk of dropping an amputated ovary within the abdomen during standing laparoscopic ovariectomies in mares. If the ovary can no longer be directly visualized with the laparoscope, the procedure is then converted to a flank laparotomy for manual retrieval of the ovary which negates the minimally invasive nature of the laparoscopic procedure. The objective was to identify if ovaries left in the abdomen after amputation would atrophy or if they re-vascularize. Standing bilateral ovariectomies were performed in mature mares, but after transection of the ovarian pedicle, the ovaries were intentionally dropped and left within the abdomen. Post-operative endocrine values were at basal levels and an improvement in all mare’s behavior and general herd dynamics was observed. While this study does not encourage to leave ovaries in the abdomen after amputation, we report no complication related to their voluntary release into the abdomen.

## 1. Introduction

Standing laparoscopic ovariectomies in sedated mares is the preferred method for ovariectomy in equids [1]. It is a safe and reliable technique, can be performed for normal or enlarged ovaries, and is associated with less morbidity and mortality than traditional approaches [1]. Dropping an amputated ovary within the abdomen during standing laparoscopic ovariectomy is a potential complication and usually occurs during manipulation at the body wall during ovary extraction from the abdomen [2,3,4,5,6,7,8]. In most circumstances, the ovary can be located laparoscopically and retrieved successfully [3,7]. However, there have been times in the authors experience and described in the literature when the ovary moves ventrally beneath intestines and can no longer be directly visualized with the laparoscope [3,4,6,7]. Since no data related to leaving an amputated ovary free in the abdomen of mares are currently available, it remains recommended to convert the procedure to a flank laparotomy with manual identification and retrieval of the dropped ovary [4,7]. The conversion to a laparotomy negates the minimally invasive nature of the laparoscopic procedure [4] and the larger incision to accommodate an arm results in the risk of incisional complications [1,9].

There is a report on 4–5 month-old fillies that describes transecting the ovarian pedicle and then leaving the ovaries within the abdominal cavity during surgery [4]. On gross examination at 10 weeks post-surgery, all ovaries showed avascular necrosis and mineralization of the deep vasculature and follicular structures [4]. The report also mentions the need for the same procedure to be repeated in mature mares with evaluation of behavioral and hormonal changes [4].

The first aim of our study was to identify if the free abdominal ovaries left within the abdomen continue to produce hormones or whether the production ceases. The second aim was to evaluate any change in behavior post-ovariectomy and to report any complications of leaving free ovaries in the abdomen. Results of this study will determine the importance in taking more invasive surgical measures to find and remove a lost amputated ovary within the abdomen or if it is in better interest for the patient to leave the ovary intra-abdominally. If there is no significant detriment to leaving an ovary in the abdomen, then more invasive surgical approaches to retrieve lost ovaries would not be indicated.

## 2. Materials and Methods

### 2.1. Cases and Presurgical Management

The study was part of a two-part study which, aside from leaving the ovaries intraabdominally, also investigated a unilateral left paralumbar fossa approach for bilateral ovariectomy. Clinical case selection bias was present and the 4 mares and 1 Molly mule enrolled in the study were from one specific ranch with the objective of improving mare behavior in order to decrease aggression between horses. The study was performed in the month of January and followed a protocol approved by the Colorado State University Institutional Animal Care and Use Committee (Protocol ID 17-7086A). Written client consent was obtained prior to participation in the study and there was no financial incentive for enrollment in the study.

Pre-operative physical examination was performed, and blood was collected for packed cell volume and total protein values in all mares. A pre-operative venous blood sample (30 mL) was obtained and placed in red-top serum tubes, allowed to clot, and then centrifuged and the serum aliquoted into a cryovial and stored at −20 °C for future endocrine panel analysis. Samples from all patients were stored frozen and the analysis was performed at the end of the study period for laboratory consistency. Feed was withheld for 12–16 h prior to surgery and mares were given access to water up to the time of surgery.

An intravenous (IV) catheter was placed in the left or right jugular vein and the mare positioned in standing stocks. Presurgical antibiotic [penicillin G procaine; AGRI-CILLIN, Norbrook Laboratories Limited, Newry, Northern Ireland, 22,000 IU/kg body weight (BW), intramuscularly (IM)] and nonsteroidal anti-inflammatory (flunixin meglumine; Prevail, VetOne, Boise, ID, USA, 1.1 mg/kg BW, IV) medications were administered approximately 30 min prior to the start of surgery. Sedation was initially achieved with detomidine hydrochloride (Dormosedan; Zoetis, Kalamazo, MI, USA, 0.01 mg/kg BW, IV) and butorphanol tartrate (Torbugesic; Zoetis, Kalamazo, MI, USA, 0.01 mg/kg BW, IV). Throughout the procedure, a level plane of sedation was maintained by a continuous infusion of detomidine hydrochloride (20 mg) in 1 L 0.9% sodium chloride (Hospira, Lake Forest, IL, USA) at a rate titrated to effect.

### 2.2. Surgical Procedure

The left paralumbar fossa was clipped with wide margins, then aseptically prepared, and draped in a routine manner. As previously described by Devick et al., the 3 determined portal sites in the left paralumbar fossa were injected subcutaneously and intramuscularly with mepivacaine hydrochloride (Carbocaine-V; Zoetis, Kalamazo, MI, USA, 0.01–0.05 mL/kg BW per site), portal incisions created, and laparoscopic cannulas (11 mm diameter, 20 cm long, Surgical Direct, DeLand, FL, USA) inserted [10]. A 30° forward viewing laparoscope (10 mm diameter, 57 cm long, Hopkins Telescope, Karl Storz Veterinary Endoscopy, Goleta, CA, USA) was used throughout the procedure and the peritoneal space was insufflated (Stryker 40 L High-Performance Insufflator, Kalamazo, MI, USA) with carbon dioxide at a variable rate up to 12 L/min and maintained at a pressure of 10–12 mmHg throughout the procedure. The left ovary was observed and mepivacaine hydrochloride (0.03–0.04 mL/kg BW) was injected along the mesovarium, mesosalpinx, and proper ligament of the ovary using a laparoscopic injection needle (5 mm diameter, 45 cm long, 19 gauge, Surgical Direct). An avascular region of the mesocolon was splash blocked with mepivacaine hydrochloride (0.01 mL/kg BW) via a laparoscopic needle and a 5–6 cm vertical incision was created through the mesocolon. The right ovary was observed through the window created in the mesocolon and mepivacaine hydrochloride (0.03–0.04 mL/kg BW) was injected into the right mesovarium, mesosalpinx, and proper ligament of the ovary. The mesosalpinx and proper ligament were transected with laparoscopic scissors just caudal to the ovary and extending 1–2 cm dorsally. The ovarian pedicle was double ligated with 4S-modified Roeder knots using USP 1 polyglyconate (Maxon, Covidien, Mansfield, MA, USA) and then the ovarian pedicle was sharply transected. The right ovary was then left in the right side of the abdominal cavity. The process was then repeated for the left ovary and the left ovarian pedicle was ligated and transected in similar fashion, and the ovary dropped in the left side of the abdomen. The incision created in the mesocolon was then apposed using a laparoscopic stapler (Multifire Endo Hernia Straight Hernaia Stapler 12–4.8 mm staples, Medtronic, Minneapolis, MN, USA). The skin of each portal incision was sutured with USP 0 polypropylene (Surgipro, Covidien, Mansfield, MA, USA) with a cruciate pattern. An aerosol bandage (AluSpray, Neogen Corporation, Lexington, KY, USA) was applied over the incision sites.

### 2.3. Post-Operative Management

The mares were monitored in hospital for 3 days post-operatively for any incisional complications such as abnormal emphysema, swelling, heat, discharge, or pain on palpation. As well, they were monitored for any change in mentation, appetite, signs of colic, fecal output, and consistency. Patients received no post-operative antibiotics and were transitioned to a course of phenylbutazone (VetriBute; VetOne, Boise, ID, USA, 2.2 mg/kg bwt orally) every 12 h for 4 days. It was recommended to confine mares to a stall or small paddock for the first 2 weeks. If there were no complications, then after the 2-week, suture removal mares were gradually returned to exercise.

At a time-point of 90 days post-operative, a venous blood sample (30 mL) was obtained and placed in red-top serum tubes, allowed to clot, and then centrifuged and the serum aliquoted into a cryovial and stored at −20 °C for future endocrine panel analysis. At 140 days post-operatively, the owner was contacted and completed a post-operative questionnaire. Owner was again contacted at 35 months post-operatively for follow up.

### 2.4. Hormone Profile Analysis

Pre- and post-operative serum samples from all mares (Table 1) were submitted to the University of California, Davis Clinical Endocrinology Laboratory and analyzed for levels of testosterone (normal reference range 20–45 pg/mL), inhibin-A (normal reference range 2–100 pg/mL), estrone sulfate (normal reference range for non-pregnant 0.1–6.0 ng/mL), anti-Müllerian hormone (AMH) (normal reference range 0.1–6.9 ng/mL), and progesterone (normal reference range for absence of active luteal tissue <0.5 ng/mL).

### 2.5. Statistical Analysis

The difference between pre- and post-operative hormone levels was calculated. A signed rank test was performed to test the significance of the difference between pre- and post-operative. SAS v9.4 (SAS Institute Inc., Cary, NC, USA) was used to perform the analysis. A *p*-value of 0.05 was set a priori to determine statistical significance.

## 3. Results

### General Results

Four mares and one Molly mule from a single ranch were admitted to Colorado State University Teaching Hospital and enrolled in the study for an elective standing bilateral ovariectomy. The owner pursued ovariectomy to improve behavior, mainly by decreasing aggression towards other horses. Mare breeds included 2 American Paints, 1 American Quarter Horse, 1 Appaloosa, and 1 Molly mule. Age was undetermined due to the unknown history of the mares but all mares were mature. The mean (range) weight was 495 kg (470–520 kg).

There were no intra-operative complications experienced in any patient. All ovaries were laparoscopically observed to be of normal size and appearance with no apparent pathologic change. Minor incisional and subcutaneous emphysema was common post-operatively but there were no other post-operative complications noted while the patients were hospitalized. Follow up with the owner was performed at 140 days post-ovariectomy. No patients experienced any complications during the 140 day post-operative period and all returned to their intended use.

Mares demonstrated a variety of pre-operative unwanted behaviors (often more than one specific behavior per mare) and included aggression towards other horses, aggression towards people, general disagreeable behavior, general excitability, problems with training, stallion like behavior, displaying constant heat, and frequent urination. Based on the owner’s evaluation at 140 day’s post-ovariectomy, all mares showed improvement in the unwanted behaviors and no signs of estrus. The owner was very satisfied with the outcome of the procedure in all mares and the Molly. At 35 months post-operatively, the owner was contacted and there were no issues or complications with the mares and Molly and all continued to do well.

Pre- and post-operative endocrine values and statistical analysis results are shown in Table 2. Testosterone values approached statistical significance (*p*-value 0.0625) in decreasing post-ovariectomy. All other hormones decreased post-ovariectomy but a significant difference was not shown.

## 4. Discussion

Standing laparoscopic ovariectomy is a safe and accepted procedure commonly performed in mares [1,2,6,11]. Performing ovariectomies laparoscopically has numerous advantages when compared with traditional surgical approaches and includes tension-free hemostasis, superior visual observation with direct observation of the ovarian pedicle for assessing hemorrhage during the procedure, reduced surgical morbidity due to the minimally invasive nature of laparoscopy, and shortened post-operative convalescent time [2,6,12,13,14,15]. The advantages of laparoscopic ovariectomies has resulted in a considerable reduction in morbidity and mortality rates compared to traditional techniques [1,3,16]. Complications are rarely encountered during laparoscopic surgery if correct technique is followed [17].

However, minimally invasive laparoscopic surgery by no means negates the potential risk of numerous complications. In general, potential complications associated with laparoscopy can be classified into complications related to general laparoscopy and those related to the specific surgical procedure being performed [17,18]. More specifically, potential complications in laparoscopic surgery include those related to patient and anesthetic selection, trocar and cannula insertion, laparoscopic instrumentation, pneumoperitoneum, and specific operative procedures [17].

Specific complications related to the standing laparoscopic ovariectomy include pain during ligation [17], and inadequate procedural hemostasis [8] such as hemorrhage as a result of slippage of the ligature from the pedicle or improper knot security [6,17,19]. Dropping an ovary within the abdomen is another reported potential complication specific to ovariectomies and usually occurs during manipulation at the body wall during ovary extraction from the abdomen [2,3,4,5,6,7,8,20]. In most circumstances the dropped ovary can be located laparoscopically and retrieved successfully [3,7,21]. However, there are times when the ovary moves ventrally beneath intestines and can no longer be directly visualized with the laparoscope [3,4,6,7]. In these cases, it is recommended to extend the incision and convert the procedure to a paralumbar celiotomy with manual identification and retrieval of the dropped ovary [4,7]. The conversion to a laparotomy negates the minimally invasive nature of the procedure [4] and a report showed all mares with post-operative flank incisional complications had larger incisions to accommodate ovaries greater than 12 cm [1].

There are numerous techniques reported to decrease the risk of dropping an ovary within the abdomen and include first to make the incision sufficiently large enough to remove ovaries without difficulty [6,8,20]. Puncturing and draining follicles intraabdominally to decrease ovarian size can be helpful for easier extraction [5]. The use of Knowles forceps properly applied and ovary exteriorization performed slowly [3] or the use of acute claw graspers and applying additional Oschner forceps on the ovary as soon as it can be seen within the body wall incision can be used to help secure the ovary during extraction [8]. Intraabdominal retrieval bags have the advantage of decreasing risk of losing the ovary [1,15], however come with an added cost to the client and are non-reusable.

Although it is common practice to leave amputated ovaries within the abdomen of heifers undergoing ovariectomy [22], the consequence of leaving an amputated ovary in the abdomen of mature mares is unknown. It is possible that leaving amputated ovaries within the abdomen would potentially result in ovarian remnant syndrome [20]. Therefore, if the ovary is dropped and cannot be located laparoscopically, it is thought to be of importance to convert the procedure to a flank laparotomy, allowing for manual retrieval of the dropped ovary [4,7]. A report on 4–5-month-old fillies describes transecting the ovarian pedicle with electrosurgical instrumentation and leaving the ovaries within the abdomen during surgery [4]. It was concluded that attachment of the ovary to the omentum and rate of neovascularization occurred at a rate incompatible with ovarian survival [4]. If ovaries become non-functional and could be left within the abdomen of mares after ovarian pedicle transection without complications, then it could reduce surgical times, technical demands, improved incisional cosmesis, and decrease post-operative incisional complications and morbidity [4]. However, in the mature mare, it is unknown if revascularization could occur rapid enough by the omentum and prevent avascular ovarian necrosis [4].

Previous studies have showed that post-operatively after ovariectomy, hormones including testosterone, inhibin, anti-mullerian hormone, and progesterone decreased to basal levels [23,24,25,26,27,28]. Results from our study showed levels of these hormones post-operatively that are very similar to a study that evaluated pre- and post-operative endocrine values in 31 mares undergoing bilateral ovariectomy with ovaries extracted [28]. These post-operative endocrine values suggest that revascularization of the ovaries did not occur in our patients. Although all hormones declined post-operatively, there were no significant differences detected between the pre- and post-operative hormone values. This lack of significance is likely due to limitations of this study which include the low case number of 5 patients. As well, the time of year that the ovariectomies were performed (January) and that the mares were in anestrus was a limitation. This time of year resulted in already low pre-operative endocrine values and resulted in a less dramatic decline between pre- and post-operative endocrine values. However, if the ovaries had revascularized by the time of the 90 day post-operative blood draw, the hormone levels would be expected to be at a higher level due to the time of year as the mares would have been cycling at that time. Since the study was performed on client mares, there was not an option for a second look laparoscopy or post-operative necropsy examination to evaluate ovarian location and ovarian histopathology to further assess ovarian viability. Other potential complications of leaving an amputated ovary intra-abdominally that were not assessed in our study include evaluation of whether there is a possibility that the ovary enveloped within the omentum could act like a mesenteric mass and cause bowel strangulation. We were not able to determine if ovarian atrophy caused some level of peritonitis predisposing to an increased risk of adhesions, yet there were no post-operative problems in these mares in the two years after the procedure. Other study limitations included client bias in the post-operative behavior change assessment and that serum was frozen and stored until the completion of the study before assays were performed. Although, preserving human sex-steroid hormones frozen has shown no detrimental effects on the hormone concentrations [29,30].

## 5. Conclusions

Leaving amputated ovaries intra-abdominally results in low post-operative endocrine levels and positive improvement in unwanted behaviors in all mares. With these results and no reported post-operative complications, it appears there was no negative effect on leaving ovaries intra-abdominally in our study population. It is the authors’ recommendation to still make every laparoscopic attempt to extract amputated ovaries during ovariectomy. Until further research is performed on a larger study population and it includes second look laparoscopy and/or post-operative necropsy and histopathology of the ovaries, it is the surgeon’s discretion whether to leave the ovary intra-abdominally or convert the procedure to a laparotomy for manual removal of the lost ovary.

## Figures and Tables

**Table 1 animals-11-00232-t001:** Pre- and post-operative endocrine results.

	TESTOSTERONE Normal Reference Range: 20–45 pg/mL		INHIBIN A Normal Reference Range: 2–60 pg/mL		AMH Normal Reference Range: 0.1–6.9 ng/mL		PROGESTERONE Reference Range for Absent Active Luteal Tissue: 0.1–0.5 ng/mL		ESTRONE SULFATE Normal Reference Range: 0.1–6.0 ng/mL	
	PREOP	POSTOP	PREOP	POSTOP	PREOP	POSTOP	PREOP	POSTOP	PREOP	POSTOP
MARE #1	36.4 pg/mL	28.4 pg/mL	16.9 pg/mL	2.22 pg/mL *	1.88 ng/mL	0.046 ng/mL	0.01 ng/mL *	0.030 ng/mL *	0.01 ng/mL *	0.01 ng/mL *
MOLLY	29.8 pg/mL	25.3 pg/mL	3.10 pg/mL *	2.85 pg/mL *	0.01 ng/mL *	0.01 ng/mL *	1.85 ng/mL	0.46 ng/mL	0.30 ng/mL	0.19 ng/mL
MARE #2	25.8 pg/mL	14.2 pg/mL	8.63 pg/mL	3.21 pg/mL *	0.99 ng/mL	0.01 ng/mL *	0.01 ng/mL *	0.01 ng/mL *	0.01 ng/mL *	0.01 ng/mL *
MARE #3	42.6 pg/mL	23.7 pg/mL	1.73 pg/mL *	4.98 pg/mL	1.70 ng/mL	0.01 ng/mL *	0.04 ng/mL *	0.01 ng/mL *	0.01 ng/mL *	0.01 ng/mL *
MARE #4	29.9 pg/mL	25.9 pg/mL	7.65 pg/mL	3.76 pg/mL *	0.17 ng/mL	0.06 ng/mL	0.02 ng/mL *	0.01 ng/mL *	0.01 ng/mL *	0.01 ng/mL *

* indicates below standard curve.

**Table 2 animals-11-00232-t002:** Pre- and post-operative endocrine results and statistical analysis.

		Range	Median	Standard Error	*p*-Value
Testosterone (pg/mL)	PRE-OP	25.8–42.6	29.9	3.0	0.063
	POST-OP	14.2–28.4	25.3	2.4	
Inhibin A (pg/mL)	PRE-OP	1.73–16.90	7.65	2.67	0.188
	POST-OP	2.22–4.98	3.21	0.47	
Amh (ng/mL)	PRE-OP	0.01–1.88	0.99	0.38	0.125
	POST-OP	0.01–0.06	0.01	0.01	
Progesterone (ng/mL)	PRE-OP	0.01–1.85	0.02	0.37	0.375
	POST-OP	0.01–0.46	0.01	0.09	
Estrone Sulfate (ng/mL)	PRE-OP	0.01–0.30	0.01	0.06	1.000
	POST-OP	0.01–0.19	0.01	0.04	

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
