# Peer review of "Effects of Leaving Amputated Ovaries Intra-Abdominally during Elective Bilateral Standing Laparoscopic Ovariectomy in Equids"

_animals, 2021, doi:10.3390/ani11010232_

Round 1

Reviewer 1 Report

Dear Authors,

In my opinion the article is acceptable in its present form.

Author Response

In my opinion the article is acceptable in its present form.

- no response required

Reviewer 2 Report

Thank you for your response.

Author Response

no response required

Reviewer 3 Report

The authors need to provide more evidence for non-functioning ovaries. In the current status, the paper is a case report. The procedures were performed in the non-breeding season. Also, the endocrine panel was analyzed during the off-season (no changes in the hormonal profile). With this hormonal profile, it is not possible to reach to any conclusion. All the values were low and they stay low, except Mare#3 which her Inhibin A actually increased! A new set of samples (during the breeding season) needs to be run to identify the ovaries' viability. There are several reports in other species that ovaries can be viable (re-vascularize) after ovariectomy. These need to be discuss in the introduction and discussion.

In addition, only five animals were tested, which makes it hard to conclude that "there was no negative effect on leaving ovaries Intra abdominally". The authors did not investigate the possibility of adhesion of ovaries to the viscera. This at least need to be discussed and be highlighted in both abstract and discussion.

How authors associated the abnormal behavior with the hormonal profile? Especially when there were no differences between pre- and post-operation concentrations?  The highest value of reported testosterone was 42pg/ml, which still is in the normal range. How authors associated this with aggression or stallion-like behaviors?

Line 17-18: “After surgery the mare’s hormone values were at low levels and an improvement in all mare’s behavior and general herd dynamics was observed”. This is not a correct statement. The values were low before the surgery too.

Author Response

Comment: The authors need to provide more evidence for non-functioning ovaries. In the current status, the paper is a case report. The procedures were performed in the non-breeding season. Also, the endocrine panel was analyzed during the off-season (no changes in the hormonal profile). With this hormonal profile, it is not possible to reach to any conclusion. All the values were low and they stay low, except Mare#3 which her Inhibin A actually increased! A new set of samples (during the breeding season) needs to be run to identify the ovaries' viability. There are several reports in other species that ovaries can be viable (re-vascularize) after ovariectomy. These need to be discuss in the introduction and discussion.

Response

  • yes agreed that the procedure was performed in non breeding season which has always been listed as a limitation of the paper
  • the post-op hormone panel was not performed in the "off-breeding season" as already mentioned in the first response to reviewers; they were performed 90 days post-op and in breeding season (April)
  • we agree that there are other species where ovaries can re-vascularize and that is what the basis for this entire study is based on  

In addition, only five animals were tested, which makes it hard to conclude that "there was no negative effect on leaving ovaries Intra abdominally". The authors did not investigate the possibility of adhesion of ovaries to the viscera. This at least need to be discussed and be highlighted in both abstract and discussion.

Response: 

  • we specifically changed the wording the last time to "it appears there was no negative effect on leaving ovaries intra-abdominally in our study population"  we are not making any bold conclusions.
  • again yes we did mention the possibility of adhesions to viscera and specifically said: "Other potential complications of leaving an amputated ovary intra-abdominally that were not assessed in our study include evaluation of if there is a possibility that the ovary enveloped within the omentum could act like a mesenteric mass and cause bowel strangulation"

How authors associated the abnormal behavior with the hormonal profile? Especially when there were no differences between pre- and post-operation concentrations?  The highest value of reported testosterone was 42pg/ml, which still is in the normal range. How authors associated this with aggression or stallion-like behaviors?

Response

  • there was no association made between the hormone profile and the behavior; the association was between ovariectomy and behavior; and there was never any stallion like behavior mentioned in the paper; these were not mares with granulosa cell tumors; it essentially eliminated the aggression between horses that is commonly observed when a mare is in estrus

Line 17-18: “After surgery the mare’s hormone values were at low levels and an improvement in all mare’s behavior and general herd dynamics was observed”. This is not a correct statement. The values were low before the surgery too.

Response

  • yes but the mares should have been cycling and have higher endocrine values if the ovaries were revascularized at the post-op analysis because it was during the 'breeding season'

This manuscript is a resubmission of an earlier submission. The following is a list of the peer review reports and author responses from that submission.

Round 1

Reviewer 1 Report

It would have been interesting to have had access to the pre-surgical diagnostic work-up. It would also have been interesting to know, whether the outcome of leaving an excised ovary intra-abdominally would have been the same regardless of the type of pathology or absence of same.

Author Response

We are happy to provide pre-surgical diagnostic work-up if wanted; they were limited to packed cell volume, total protein, physical exam, and the endocrine values that have been added to the manuscript

it has been added to the manuscript that all ovaries were visualized and none had evidence of pathologic change and no endocrine panels indicated pathologic change of the ovaries

Reviewer 2 Report

Dear Authors,

In general the manuscript "Effects of leaving amputated ovaries intra-abdominally during elective bilateral standing laparoscopic ovariectomy in equids" is well written. The obtained data provide a support for the veterinarians.  I`m sending my suggestions for the improvement of the manuscript

  • Affiliation: page 1 line 8: "Affiliation 2; [email protected]" please remove this additional affiliation.
  • Materials and methods: page 2 line 69-70: "The study was performed in the month of January and followed a protocol approved by the Colorado State University Institutional Animal Care and Use Committee." - please add the number of this permission.

Author Response

the following line was not in the manuscript submitted but has been removed:

  • page 1 line 8: "Affiliation 2; [email protected]" please remove this additional affiliation. 

page 2 line 69-70: "The study was performed in the month of January and followed a protocol approved by the Colorado State University Institutional Animal Care and Use Committee." - please add the number of this permission.

  • the protocol ID has been added to the manuscript

Reviewer 3 Report

An interesting study where benefit to the welfare of mares undergoing the procedure can be clearly identified.

A few minor grammar mistakes:

19 - mares, mare's

224 - 'do' changed to 'due'

225 - 'where' changed to 'were' 

Author Response

the following changes have been made to the manuscript

  • 19 - mares, mare's

  • 224 - 'do' changed to 'due'

  • 225 - 'where' changed to 'were' 

Reviewer 4 Report

Dear Dr. Devick and Dr. Hendrickson, The question "what to do when dropping an ovary during laparoscopy" is interesting to anyone who does laparoscopy. This is therefore a very important study. However, there are limitations to this study that I struggle with: 1. If this farm is specialized on improving the behavior of mares (line 68), how can you assume that the improved behavior 140 days after surgery is not simply the result of training? Is there a way to show that the improvement seen in these horses/mule is any different than in the mares they have on their farm and did not undergo ovariectomy? Were the behavioral problems in these mares any different than in the other mares they had on their farm - i.e. why did they chose these horses/mule for the study? 2. Clarification of the point above is important, because (as shown in your study in the Canadian Veterinary Journal in 2020), hormone levels do not correlate with undesirable behavior. So the decrease in hormone levels itself is not proof that the surgery had anything to do with the changes in behavior. Assuming that behavioral change is the consequence of ovariectomy, only really works if nothing else in the environment of the horse changes. So perhaps these mares had already been on that farm for months or years and this was the last option? 3. The lack of significant changes in hormone levels is not helping your study, although your explanation does make sense. Would it be possible to re-sample the horses in January 2021 or would that be a waste of time, since there should not be any fluctuations in hormone levels anymore. Or could you use the lack of changes in these hormone levels throughout a year/over the course of several months (re-sampling on several occasions) to show that the ovaries are inactive? 4. Could you please add reference ranges from the lab to the manuscript? Are reference ranges for mules available or do we assume that they are the same? I think it would be helpful to show hormone levels for each individual horse and the mule - that should be simple with five animals in the study. 5. Please define "basal levels" - are you referring to levels during anestrus? Since pre-operative blood samples were taken during anestrus, all levels (pre- and post sx) should be at basal levels, but you are using the term only to talk about post-op levels. I might be missing something here and would appreciate clarification. 6. Show the pre- and postoperative hormone levels of the normal horses from your study in CVJ 2020 for comparison. They are going to be different, but comparison should be made easy for the reader. 7. Did the ovaries in all mares appear normal during laparoscopy? Did you do any ultrasound exam of the ovaries (I assume not, but wanted to double-check)? Thank you.

Author Response

If this farm is specialized on improving the behavior of mares (line 68), how can you assume that the improved behavior 140 days after surgery is not simply the result of training? Is there a way to show that the improvement seen in these horses/mule is any different than in the mares they have on their farm and did not undergo ovariectomy? Were the behavioral problems in these mares any different than in the other mares they had on their farm - i.e. why did they chose these horses/mule for the study? 

  • have changed the wording of the sentence as it was miss understood. The ranch does not specialize in improving behavior but they are actually a dude ranch with numerous horses for guests to ride. So they elect to have mares spayed that show aggressive behavior to other horses and this way they have less issues housing horses together in paddocks
  • the mares behavior post-op is only compared to its pre-op behavior; there were no other non-spayed mares on the place

2. Clarification of the point above is important, because (as shown in your study in the Canadian Veterinary Journal in 2020), hormone levels do not correlate with undesirable behavior. So the decrease in hormone levels itself is not proof that the surgery had anything to do with the changes in behavior.

  • yes that is correct that no specific elevated hormones were correlated with an undesirable behavior in that study but in general 89% owners were satisfied with the outcome in the mares behavior improved post-OVX

Assuming that behavioral change is the consequence of ovariectomy, only really works if nothing else in the environment of the horse changes. So perhaps these mares had already been on that farm for months or years and this was the last option?

  • correct; if the mares are noticed to be aggressive to other horses and disrupt the heard then they are spayed and thus act more like the geldings 

3. The lack of significant changes in hormone levels is not helping your study, although your explanation does make sense. Would it be possible to re-sample the horses in January 2021 or would that be a waste of time, since there should not be any fluctuations in hormone levels anymore. Or could you use the lack of changes in these hormone levels throughout a year/over the course of several months (re-sampling on several occasions) to show that the ovaries are inactive?

  • yes the time of year that we did the study was poor study design in hindsight; we do not have anymore funding for the project and repeated endocrine analysis is not affordable at this time

4. Could you please add reference ranges from the lab to the manuscript? Are reference ranges for mules available or do we assume that they are the same? I think it would be helpful to show hormone levels for each individual horse and the mule - that should be simple with five animals in the study.

  • reference ranges have been added
  • hormone levels for all 5 patients have been added

5. Please define "basal levels" - are you referring to levels during anestrus? Since pre-operative blood samples were taken during anestrus, all levels (pre- and post sx) should be at basal levels, but you are using the term only to talk about post-op levels. I might be missing something here and would appreciate clarification. 

  • yes, basal by definition means belonging to a bottom or base
  • I understand the confusion and the word basal has been removed from throughout the manuscript except the one spot that references other studies
  • yes I agree that a mare in anestrus would also be considered to have hormones at basal levels and I don't think we ever said they were not; the median values for the hormones post-op were even lower than pre-op however not significant likely since such low case numbers

6. Show the pre- and postoperative hormone levels of the normal horses from your study in CVJ 2020 for comparison. They are going to be different, but comparison should be made easy for the reader.

  • I can check with CVJ but think that would be a breech of copyright agreement to use results from that study; this study and 5 other papers have been referenced for readers to find
7. Did the ovaries in all mares appear normal during laparoscopy? Did you do any ultrasound exam of the ovaries (I assume not, but wanted to double-check)?
  • yes all appeared normal; no ultrasound exams

Reviewer 5 Report

The authors present a study of 5 equids that underwent laparoscopic transection of the ovarian blood supply and pedicle without removal of the ovary. No complications were reported. Based on these 5 cases, they conclude that it is OK to leave the ovary in place if dropped. A larger study of 10, 4-5 month old foals did the same thing but completed a post-mortem exam found that many of the ovaries were adhered to the ventral omentum but did not vascularize, with several ovaries freely moving in the abdominal cavity. My largest concern is that the ovaries attached to the omentum may develop a pedicle and act similar to a lipoma and ensnare a piece of bowel.

I think that the authors are making a bold statement that has little factual support. The basic conclusion that can made after reading this paper and the previous one is that in 15 cases of intentionally leaving the ovaries in place, no complications were noted except omental adhesion. The follow up time was limited. This paper does nothing to change my understanding the consequences of leaving free ovaries after laparoscopic transection.

Author Response

Based on these 5 cases, they conclude that it is OK to leave the ovary in place if dropped. A larger study of 10, 4-5 month old foals did the same thing but completed a post-mortem exam found that many of the ovaries were adhered to the ventral omentum but did not vascularize, with several ovaries freely moving in the abdominal cavity. My largest concern is that the ovaries attached to the omentum may develop a pedicle and act similar to a lipoma and ensnare a piece of bowel.

  • yes we agree with this statement and mentioned this exact risk in the limitation section (see line 233-235 in initial manuscript)

I think that the authors are making a bold statement that has little factual support.

  • there was no recommendation made to leave ovaries in the abdomen; the statement that no negative effects were observed is true and is actually factual; however I do agree that the case numbers are low and conclusions shouldn't be made until larger study with second look or necropsy performed; this small population size has low power and we reported the findings to hopefully stimulate further studies with significant conclusions

The basic conclusion that can made after reading this paper and the previous one is that in 15 cases of intentionally leaving the ovaries in place, no complications were noted except omental adhesion.

  • correct; this study with low case numbers is only going to have basic conclusions and is meant as more of a pilot study leading to one with higher case number, second look/necropsy and histo ect.
  • the main difference and additional information between the foal study and our study is the endocrine panels and that the foal ovaries were obviously never actively cycling and we though may be less likely to revascularize than in a mature mare

The follow up time was limited. 

  • it has now been 35 months since the study performed; owner has been contacted and manuscript updated

Reviewer 6 Report

The authors need to provide more evidence for non-functioning ovaries. The procedures were performed in the non-breeding season. Also, the endocrine panel was analyzed during the off-season (no changes in the hormonal profile). A new set of samples (during the breeding season) needs to be run to identify the ovaries' viability. There are reports in other species that ovaries can be viable after ovariectomy. In dogs, autografted ovaries survived and produced oocytes.

In addition, only five animals were tested, which makes it hard to conclude that "there is no negative effect on leaving ovaries Intra abdominally". The authors also need to discuss the possibility of adhesion of ovaries to the viscera and its possible effects.

How authors associated the abnormal behavior with the hormonal profile? Especially when there were no differences between pre- and post-operation concentrations? The highest value of reported testosterone was 42pg/ml, which still is in the normal range. How authors associated this with aggression or stallion-like behaviors?

Minor:
Please provide the hormonal concentration and their dynamic for each individual separately (perhaps line graph).
Please provide information regarding the sensitivity of the tests.

Author Response

The authors need to provide more evidence for non-functioning ovaries. The procedures were performed in the non-breeding season. Also, the endocrine panel was analyzed during the off-season (no changes in the hormonal profile). A new set of samples (during the breeding season) needs to be run to identify the ovaries' viability.

  • the post-op sample was was actually taken during breeding season and mares should have been cycling (April)

There are reports in other species that ovaries can be viable after ovariectomy. In dogs, autografted ovaries survived and produced oocytes.

  • yes we agree with the statement and that is essentially the purpose of our study to see if there was any indication of ovaries revascularizing and being functional

In addition, only five animals were tested, which makes it hard to conclude that "there is no negative effect on leaving ovaries Intra abdominally". 

  • yes this study population has low case numbers; however we believe it is appropriate to conclude that there were no negative effects in the 5 horses in our study; readers will easily identify that the case numbers are low and results are not significant but more reported; the conclusion has been reworded though to say there were no complications in our study population to make it more clear and not try to over portray the findings

The authors also need to discuss the possibility of adhesion of ovaries to the viscera and its possible effects.

  • this has been discussed in the original manuscript lines 233-235

How authors associated the abnormal behavior with the hormonal profile? Especially when there were no differences between pre- and post-operation concentrations?

  • we did not provide any correlation with specific hormones to abnormal behavior; but in general there was a general relation between OVX and change in abnormal behavior

The highest value of reported testosterone was 42pg/ml, which still is in the normal range. How authors associated this with aggression or stallion-like behaviors?

  • we did not associate these together and no mares in the study had stallion like behavior; or any suspicion of GCT; the behaviors observed pre-op was more characteristic of behaviors that are exhibited during estrus which is the aggression towards other horses that the owners wanted to improve

Please provide the hormonal concentration and their dynamic for each individual separately (perhaps line graph).

  • a table of all mares hormones has been added as per another reviewer request

Please provide information regarding the sensitivity of the tests.

  • reference ranges have been added